# The NRF2/KEAP1 Axis in the Regulation of Tumor Metabolism: Mechanisms and Therapeutic Perspectives

**DOI:** 10.3390/biom10050791

**Published:** 2020-05-20

**Authors:** Emiliano Panieri, Pelin Telkoparan-Akillilar, Sibel Suzen, Luciano Saso

**Affiliations:** 1Department of Physiology and Pharmacology “Vittorio Erspamer”, Sapienza University of Rome, 00185 Rome, Italy; 2Department of Medical Biology, Faculty of Medicine, Yuksek Ihtisas University, 06520 Balgat, Ankara, Turkey; pelinta@yiu.edu.tr; 3Department of Pharmaceutical Chemistry, Faculty of Pharmacy, Ankara University, 06100 Tandogan, Ankara, Turkey; sibel.suzen@pharmacy.ankara.edu.tr

**Keywords:** NRF2–KEAP1, oxidative stress, metabolic reprogramming, reactive oxygen species, chemoresistance

## Abstract

The NRF2/KEAP1 pathway is a fundamental signaling cascade that controls multiple cytoprotective responses through the induction of a complex transcriptional program that ultimately renders cancer cells resistant to oxidative, metabolic and therapeutic stress. Interestingly, accumulating evidence in recent years has indicated that metabolic reprogramming is closely interrelated with the regulation of redox homeostasis, suggesting that the disruption of NRF2 signaling might represent a valid therapeutic strategy against a variety of solid and hematologic cancers. These aspects will be the focus of the present review.

## 1. Introduction

NRF2 is a master regulator of the cytoprotective responses induced by xenobiotic, oxidative or metabolic stress and is strongly implicated in cancer cells’ metabolic reprogramming, along with multiple oncogenic pathways. Indeed, NRF2 controls the expression and function of key metabolic enzymes belonging to the glucose and glutamine catabolism, pentose phosphate pathway, glycolysis, as well as the folate or Krebs cycle with the dual goal of supporting biosynthetic processes and redox homeostasis through the constant supply of NADPH (nicotinamide adenine dinucleotide phosphate, in its reduced form). On the other hand, NRF2 can also strongly activate the uptake of amino acids used for the synthesis of GSH or modulate the expression of iron homeostasis regulators, ultimately influencing the susceptibility to ferroptosis. It is noteworthy that since NRF2-driven alterations of cancer metabolism and growth are frequently found in malignant tumors of different origin, the strong dependence on this signaling pathway, especially in NRF2-addicted cancer cells, can also reveal specific vulnerabilities that might be therapeutically exploited. In this review, we will describe the components of the NRF2/KEAP1 pathway, its role in the regulation of metabolic rewiring and the potential strategies to overcome its activation by targeting the main effectors of its pro-oncogenic function. 

## 2. NRF2 and KEAP1 Structure and Function

### 2.1. NRF2 Structure 

NRF2 is an important transcription factor encoded by the nuclear factor, erythroid derivative 2 like 2 (*NFE2L2*) gene and plays a critical role in the management of oxidative and electrophilic stress [1]. The human NRF2 protein consists of 605 amino acids and has an estimated weight of ~66 KDa [2]. NRF2 is a member of the Cap’n’collar (CNC) basic leucine zipper (bZIP) transcription factor family that entails members having a conserved 43 amino acid homology region called the CNC, located at the N-terminal DNA-binding domain [3]. NRF2 consists of seven highly conserved domains known as the NRF2 erythroid-derived CNC homology protein homology domains (Neh 1–Neh7) (Figure 1A) [4,5]. The Neh1 domain contains a bZIP DNA-binding motif that mediates the heterodimerization with other bZip-containing proteins including small MAF proteins (MAFF, MAFG, MAFK) [6]. The highly conserved Neh2 domain is located in the N-terminal region of NRF2 and binds to its negative regulator Kelch-like ECH-related protein 1 (KEAP1) via DLG and ETGE motifs [7]. The Neh3 domain is located in the C-terminal region of the NRF2 protein and facilitates NRF2 activation through interaction with chromo-ATPase/helicase DNA-binding protein 6 (CHD6) [8]. Neh4 and Neh5 domains are also critical for the transcriptional activity of NRF2 and mediate the interaction with the cyclic AMP response element-binding protein (CREB)-binding protein (CBP) and receptor-associated coactivator (RAC) [9]. The Neh6 domain includes two binding sites (DSGIS and DSAPGS motifs) that are involved in the KEAP1-indepedent regulation of NRF2 via the β-transducin repeat-containing protein (β-TrCP) and GSK-3β phosphorylation [10,11]. The seventh domain is known as Neh7 and interacts with the nuclear receptor retinoic X receptor alpha (RXRa), inhibiting NRF2 target genes expression [12].

### 2.2. KEAP1 Structure

Kelch-like ECH-associated protein 1 (KEAP1) comprises 624 amino acids with a molecular weight of 69 kDa. KEAP1 interacts with NRF2, binding to the Neh2 domain as a homodimer and controls the stability of NRF2. KEAP1 belongs to the Broad complex Tramtrack and Bric-à-Brac (BTB)-Kelch family of proteins and these family members assemble with the Cul3-containing E3 ubiquitin ligase complex that induces the proteasomal degradation of target proteins [13,14]. KEAP1 consists of five domains (see Figure 1B); N-terminal region (NTR), the Broad complex Tramtrack and Bric-à-Brac (BTB) domain, the intervening region (IVR), the Kelch domain/double glycine repeat (DGR) and the C-terminal region (CTR) [15]. The N-terminal BTB domain mediates KEAP1 homodimerization and the Cul3-E3-ligase binding that is critical for ubiquitination and the proteasomal degradation of NRF2. Additionally, the BTB domain contains cysteine residue Cys151 that is thought to sense oxidative stress levels [16]. Similar to the BTB domain, the IVR domain consists of reactive cysteine residues; Cys257, Cys273, Cys288 and Cys297 that are particularly susceptible to reactive oxygen species (ROS)-induced modifications and promote KEAP1-dependent NRF2 ubiquitination [17]. The C-terminal Kelch domain includes six Kelch-repeats that mediate the binding of KEAP1 to the ETGE or DLG motifs located within the Neh2 domain of NRF2 [18]. The structural features of NRF2 and KEAP1 are reported in Figure 1.

### 2.3. NRF2 and KEAP1 Regulation under Normal and Stressed Conditions 

KEAP1 is one of the key regulators of NRF2 protein stability. Under unstressed homeostatic conditions, NRF2 localizes in the cytosol and binds to KEAP1 which is a substrate of Cullin 3-based ubiquitin E3 ligase complex (Cul3) and facilitates NRF2 ubiquitination. KEAP1 therefore controls NRF2 stability in the cytosol through the proteasome degradation pathway. Within the cells, KEAP1 exists in the form of homodimers interacting through their BTB domains and facilitates NRF2 recognition by the Cul3-dependent ubiquitin ligase (E3) complex by interacting with the ETGE and DLG motifs of the Neh2 domain within NRF2 [19]. This interaction promotes NRF2 ubiquitination and rapid proteasomal degradation [20]. This mechanism ensures that under basal conditions NRF2 is maintained at low levels, avoiding the unnecessary expression of its target genes.

In marked contrast, the exposure to reactive oxygen species (ROS), electrophiles or other stressors, induces a conformational change in the KEAP1/Cul3/Rbx/NRF2 complex through the modification of the Cys151, Cys273 or Cys278 residues located in the BTB and IVR domains of KEAP1 [7,17]. As a consequence, NRF2 dissociates from KEAP1 and translocates into the nucleus, wherein it interacts with small MAF proteins (sMAFs) and other partners. The heterodimers subsequently bind to the antioxidant responsive elements (AREs) located in the promoter of NRF2 target genes and induce the expression of a battery of cytoprotective genes [5,21,22]. The mechanism of canonical NRF2 activation is illustrated in Figure 2. Alongside with the KEAP1-dependent regulation, additional mechanisms controlling NRF2 stability have been revealed. Among them, the β-TrCP–SKP1–RBX1–CUL1 E3 ubiquitin ligase complex primes NRF2 for proteasomal degradation upon GSK-3β-dependent phosphorylation within the Neh6 domain of NRF2, which is inhibited by PI3K/AKT activation [11,23]. In other cases, specific interactors such as p62 [24], DPP3 [25,26] or p21 [27] can compete with KEAP1 for NRF2 binding and prevent its degradation. These mechanisms are collectively referred to non-canonical NRF2 regulatory pathways. 

### 2.4. The Transcriptional Program Elicited by NRF2 Activation and Its Biological Effects

NRF2 coordinates cellular defense mechanisms against oxidants and xenobiotics by regulating the expression of more than 500 genes codying for antioxidants, detoxification or metabolic enzymes and multi drug resistance-associated protein transporters [28,29]. NRF2 plays a key role in controlling the levels and redox status of glutathione (GSH) by directly controlling the expression of the two subunits of the glutamate–cysteine ligase (GCL) complex (*GCLC* and *GCLM*) involved in GSH synthesis and the enzyme glutathione reductase (GSR) involved in GSH regeneration [30]. In addition, NRF2 transcriptionally regulates several ROS-detoxifying enzymes including glutathione peroxidase 2 (*GPX2*) and glutathione S-transferases (*GSTA1,2,3,5, GSTM* 1-3 and *GSTP1*) [31,32] or the thioredoxin antioxidant system by controlling the expression of thioredoxin (*TRX*) and *TXNIP* (thioredoxin-inhibitor) [33,34,35]. NRF2 also controls NADPH levels by influencing the expression of NADPH-generating enzymes such as glucose-6-phosphate dehydrogenase (*G6PD*), isocitrate dehydrogenase 1 (*IDH1*), and malic enzyme 1 (*ME1*) [36]. Another important function of NRF2 is providing important contributions to iron signaling by regulating the expression of *HMOX1* (heme oxygenase-1), which is an important enzyme that catalyzes the conversion of heme to biliverdin [37]. Moreover, NRF2 has been shown to play a major role in xenobiotics and drug detoxification by regulating the expression of phase-I and phase-II drug-metabolizing enzymes as well as phase-III drug transport proteins [38,39]. In summary, NRF2 is a master regulator of the cellular response against xenobiotics and oxidative stress but recent advances in genome-wide association studies (GWASs) have indicated that its role goes well beyond the regulation of the antioxidant response and drug metabolism. Indeed, NRF2 prevents the intracellular accumulation of abnormal proteins by regulating unfolded protein response-related gene expression and the proteasomal degradation of misfolded/unfolded proteins [40,41]. Furthermore, the chemical activation of NRF2 modifies the transcriptional activation of circadian genes [42] and NRF2 is also effective in other critical cellular processes such as stem cell self-renewal [43], embryonic stem cell differentiation [44], inflammatory response [45], proliferation [37,46,47], autophagy [48], apoptosis [49] and metastasis [46]. A list of some NRF2 target genes is shown in Table 1.

## 3. Role of the NRF2/KEAP1 Pathway in Tumor Metabolism

It is becoming increasingly clear that pro-oncogenic alterations of the NRF2/KEAP1 pathway play a crucial role in driving the metabolic rewiring of cancer cells, orchestrating a multi-layered transcriptional program that ultimately provides the precursor molecules to support cancer cell proliferation and the reducing equivalents to cope with the augmented bulk of intracellular ROS resulting from the malignant progression. In the next sections, we will describe the role of NRF2 in the regulation of tumor metabolism with a particular emphasis on the interconnection between some metabolic processes and the control of tumor redox homeostasis. 

### 3.1. NRF2 Controls Mitochondrial Function Linking Metabolism to Redox Balance 

Mitochondria are crucial organelles primarily involved in ATP synthesis but also regulating a vast array of cellular processes including the tricarboxylic acid (TCA) cycle, fatty acids and amino acids metabolism, calcium and ROS homeostasis as well as cellular apoptosis. It is noteworthy that NRF2 can influence the mitochondrial metabolism at multiple levels, controlling substrate availability or the flux rate into the mitochondrial electron transport chain (mETC) but also the dynamics of mitochondria fission/fusion, the clearance of damaged mitochondria (mitophagy) and the mitochondrial biogenesis [65,66]. In a similar way, NRF2 can also control the redox status of the mitochondrial GSH pool and the expression of antioxidant enzymes with mitochondrial localization.

#### 3.1.1. NRF2 Regulates Mitochondrial Biogenesis, Turnover and Mitochondrial Network Dynamics

It is known that mitochondrial dysfunction is a common hallmark of cancer cells potentially caused by metabolic rewiring, altered redox balance, deregulated fission/fusion dynamics or mitochondrial turnover [67]. In this context, NRF2 can induce tumor adaptation to adverse conditions, further promoting tumorigenesis. In this respect, by using a panel of prostate (DU145), osteosarcoma (U2OS) and breast (MCF-7) cancer cells, Riis et al. showed that NRF2 was a key player in downstream IGF-1 (insulin-like growth factor 1) signaling and required for the IGF-1-dependent induction of BNIP3 (BCL2/adenovirus E1B 19 kDa protein-interacting protein 3), a regulator of cell apoptosis and mitophagy. Mechanistically, IGF-1 prevented NRF2 degradation by enhancing the PI3K/AKT-mediated phosphorylation of GSK-3β, thereby facilitating the NRF2 nuclear accumulation and subsequent BNIP3 induction. It is noteworthy that *NRF2* silencing strongly altered the mitochondrial morphology, biogenesis and turnover, indicating that NRF2 is a key effector of the IGF-1 pathway connecting cell growth to mitochondrial homeostasis and cell survival in cancer [68]. The first evidence of a link between NRF2 and mitochondrial biogenesis came from Piantadosi et al., who firstly proved that HO-1 prevented cardiomyocyte apoptosis through AKT activation and subsequent NRF2-driven induction of *NRF-1/alpha-PAL* genes, which in turn also enhanced *ND1* (NADH dehydrogenase subunit 1) and *COX* (cytochrome-*c* oxidase subunit 1) expression, promoting mitochondrial biogenesis [69]. Subsequently, the natural compound resveratrol was found to induce mitochondrial biogenesis in Lipopolysaccharides (LPS)-treated HepG2 hepatocellular cancer cells (HCC) through the sequential production of Nitric and Carbon oxide, the latter event requiring AKT phosphorylation and NRF2-dependent HO-1 induction [70]. These observations were further substantiated in a later study wherein the treatment of colon cancer (CC) cells (SW480, HT29, and HCT116) with the aldose reductase inhibitor fidarestat, EGF (epithelial growth factor) or their combination, was seen to markedly increase the NRF2 nuclear accumulation and the transcriptional activity also enhancing the protein contents of NQO1 and HO-1. Mechanistically, the combined use of EGF and fidarestat promoted the mitochondrial biogenesis and prevented mitochondrial DNA damage under stress conditions by inducing AMPK-α1 activation, an event presumably responsible for the NRF2 phosphorylation and subsequent overexpression of *PGC-1α*, *NRF1* and *TFAM* genes, despite the fact that a causal role for NRF2 was not formally proven [71]. Finally, another inducer of NRF2, the isothiocyanate sulforaphane (SFN) was seen to promote mitochondrial biogenesis in PC3 prostate cancer (PC) and LLCPK1 normal renal epithelial cells through the enhanced expression of TFAM, MT-ND1, and NRF1 proteins. Intriguingly however, SFN modulated in opposite ways the dynamics of the mitochondrial network, promoting the fusion in LLCPK1 and fission in PC3 cancer cells, respectively leading to the activation of intrinsic apoptosis or the induction of cytoprotective mechanisms. Moreover, in this case, despite both NRF1 and TFAM being regarded as NRF2 target genes, it is unclear to which extent and how NRF2 is implicated in the differential modulation of the mitochondrial dynamics in PC3 and LLCPK1 cells [72]. It must also be emphasized that the use of NRF2 activators such as SFN has been associated with off-target effects including the derepression of long terminal repeats, suggesting that NRF2 modulation might have both beneficial as well as detrimental effects, in agreement with the concept of a hormetic response [73]. Another important process for the quality control of mitochondrial integrity and homeostasis is represented by mitophagy, wherein damaged mitochondria are sequestered in autophagosomes and subsequently degraded by the lysosomal compartment. In this respect, several studies have indicated that NRF2 plays an important role in regulating mitophagy in different cancer cells. For instance, by exposing SH-SY5Y neuroblastoma (NB) cells to oxidative stress (OS) conditions, Murata et al. showed that NRF2 could directly enhance the transcription of *PINK1* mRNA (PTEN-induced kinase 1), an essential regulator of mitochondrial quality control that promoted cell survival by mediating the recognition of damaged mitochondria [74]. Other data have suggested that PINK1 might also act upstream NRF2 since the induction of mitophagy in SH-SY5Y cells by mitochondrial uncoupling was found to increase the mRNA and protein levels of the autophagy regulator p62/SQSTM1, and the lysosomal enzyme glucocerebrosidase (GCase) was induced by NRF2 nuclear translocation but hampered by *PINK1* silencing [75]. Lastly, in a recent study it was shown that the triterpenes ursolic and oleanolic acids could induce mitophagy in A549 non-small cell lung cancer (NSCLC) cells by promoting PINK1 upregulation and its recruitment to the outer mitochondrial membrane, presumably in response to the increased NRF2 expression caused by ROS overproduction, despite the lack of mechanistic evidence that precludes further speculation [76]. Taken together, these data suggest that the induction of mitophagy represents an important mechanism through which NRF2 can support cancer cell survival under stress conditions dictated by metabolic or redox changes. 

#### 3.1.2. NRF2 Regulates Mitochondrial Respiration and Redox Homeostasis

The direct regulation of mitochondrial respiratory complexes by NRF2 has been identified in different experimental systems. For example, Yan et al. have shown that the phenolic compound punicalagin was able to attenuate the loss of mitochondrial membrane potential (MMP) and the ATP depletion caused by palmitate overload in HepG2 cells through the induction of ERK phosphorylation and subsequent NRF2 nuclear accumulation. It is however unclear whether NRF2 nuclear translocation could also restore the mETC activity or just prevent the MMP drop [77]. Consistently, other data from SH-SY5Y NB cells have indicated that lipoic acid, a mitochondrial cofactor, could promote NRF2 nuclear translocation and stimulate the expression of antioxidant genes and mETC components such as ND1 (NADH: ubiquinone oxidoreductase) and COX2 (cytochrome-C oxidase subunit II), preventing the ATP depletion induced by acrylamide and restoring the mitochondrial membrane potential (MMP) [78]. Finally, recent work has shown that *NRF2* silencing in HT29 and HCT116 CC cells produced a strong decrease in the levels of MT-CO1 (mitochondria-encoded cytochrome c oxidase subunit-1), a component of the mETC complex IV, causing the loss of MMP, decreased O_2_ consumption, ATP depletion and AMPKα activation. The same mechanism was validated also in MCF-7 and MDA-MB-231 breast cancer (BC) cells and in vivo xenografts derived from *NRF2*-silenced HT29 cells [79]. It has been proposed that NRF2 might also regulate the mitochondrial bioenergetics by directly controlling the availability of respiration substrates in the mitochondria, as evidenced in primary murine neurons and mouse embryonic fibroblasts (MEFs) [80], despite that these data have not been confirmed for cancer cells. However, another study from Kim et al. used HCT116 and HT29 colorectal cancer (CRC) cells with stable *NRF2* knockdown to prove that its deficiency hampered ATP production and O_2_ consumption under hypoxia through yet unknown mechanisms, suppressing tumor angiogenesis in vivo [81]. The crosstalk between mitochondria and NRF2 is well established in terms of redox regulation. Indeed, NRF2 shapes the redox status of the mitochondrial GSH pool by promoting the expression of the enzymes involved in GSH synthesis [82,83,84] or NADPH production (i.e., G6PD, 6PGD, TKT) [30,85,86] as well as mitochondrial antioxidant enzymes such as SOD-2, GPX1, GPX4, PRDX3, PRDX5, TRX2, and TRXR2 [87,88,89,90]. Experimental work from Kovac et al. provided an explanation on how NRF2 might control compartment-specific redox balance since in a mouse model with graded NRF2 expression, this transcription factor was seen to regulate the mRNA levels of the cytosolic NOX2 and the mitochondrial NOX4 isoforms, despite the fact that it is unknown whether the same also occurs in cancer cells [91]. Another study from Bao et al. revealed that NRF2 expression was indirectly regulated by TRMP2 (Transient receptor potential melastatin channel subfamily member 2), an ion channel frequently overexpressed in cancer that promotes mitochondrial function counteracting OS. Here, SH-SY5Y NB cells and xenografts with genetic ablation of *TRPM2* exhibited a marked decrease in the cytosolic and nuclear content of both NRF2 and IQGAP1, which is a protein that increases NRF2 stability and activation through a calcium-dependent process. As a consequence, *TRPM2*-KO cells were characterized by GSH, NADPH, NADH, GTP and ATP depletion due to the impaired expression of NRF2-inducible enzymes with metabolic (GLS, MTHFD2) or antioxidant (GCLC, GCLM, GSS) functions and decreased cell viability, a phenotype that was at least in part reverted by NRF2 reconstitution [92]. In conclusion these data suggested that NRF2 can maintain not only the mitochondrial redox homeostasis but also contribute to redox-independent functions including mitochondrial dynamics, turnover and biogenesis, expanding the list of its regulatory processes, a phenomenon that can at least in part account for the divergent effects observed in cancer or normal cells [93]. It should be however emphasized that NRF2 activation might also represent a consequence rather than a cause of mitochondrial alterations, so its precise role must to be carefully determined in each experimental model.

### 3.2. NRF2 Regulates Fatty Acids Metabolism 

As for lipids metabolism, NRF2 can positively regulate catabolic but conversely suppress anabolic processes in MEFs (mouse embryonic fibroblasts), isolated mitochondria and mice models since its constitutive expression enhanced both fatty acids oxidation (FAO) and mitochondrial respiration, while the opposite occurred in the presence of *NRF2*-KO [94]. Consistently, Pang et al. used HEK-293T to reveal that NRF2 could control FAO in distinct subcellular compartments by modulating the expression of the carnitine palmitoyltransferase isoforms (*CPT1*, *CPT2*) within mitochondria but also acyl-CoA oxidase 1 and 2 (*ACOX1*, *ACOX2*), two peroxisomal enzymes implicated in lipids beta-oxidation [95]. On the other hand, it is well recognized that NRF2 suppresses lipid biosynthesis through multiple ways, and thereby this might decrease NADPH consumption in cancer cells to support antioxidant systems. In this regard, by using murine models expressing different levels of NRF2, Wu et al. showed that the hepatic mRNA levels of the enzymes fatty acid synthase (*Fasn*), fatty acid desaturase (*Fads1*, *Fads2*), stearoyl-CoA desaturase (*Scd1*), fatty acid elongases (*Elovl2,3,5,6* and *Cyb5r3*) and ATP-citrate lyase (*Acly*), were induced in *NRF2*-null mice and conversely suppressed in *KEAP1*-KO mice. However, so far these observations have not been validated in cancer models. By contrast, NRF2 was found to transcriptionally induce FAO genes and lipases promoting the degradation of damaged lipids [86], a mechanism that might provide reducing power in the form of NADPH also in cancer cells, since CPT1 inhibition by etomoxir in SF188 glioblastoma (GBM) cells was found to markedly deplete the ATP and NADPH levels, inducing ROS accumulation [96]. Taken together, these studies have indicated that the NRF2 pathway can represent the molecular link between metabolic processes controlling the lipid metabolism or mitochondrial function and the control of redox homeostasis in malignant cells.

### 3.3. NRF2 Regulates Aminoacids Metabolism 

#### 3.3.1. NRF2 Controls Aminoacids Uptake and Biosynthesis to Support Proliferation and Survival

Accumulating evidence has indicated that NRF2 can participate in the regulation of the intracellular pool of aminoacids by influencing the molecular pathways involved in their uptake or biosynthesis, contributing to sustaining cancer cell proliferation, metabolic rewiring and redox balance. In this respect, several studies have focused on the crosstalk between NRF2 and ATF4, a transcription factor induced under oxidative, metabolic and ER stress. In a seminal study from De Nicola et al., metabolic tracing and transcriptional profiling on a panel of NSCLC cell lines revealed that NRF2 controlled the transcription of *PHGDH*, *PSAT1* and *SHMT2*, the key enzymes involved in serine/glycine biosynthesis through ATF4 activation. It is noteworthy that not only this event was necessary to support both nucleotide and GSH synthesis but the high expression of those genes also identified the lung cancer patients with a poorer prognosis [97]. Interestingly, by using NSCLC cells subdued to nutrient stress, Gwinn et al. found that the activation of the KRAS/AKT/NRF2/ATF4 axis enhanced the expression of the *SLC1A5, SLC38A2, SLC7A5, SLC7A1* and *SLC7A11* genes, respectively codying for the amino acid transporters ASCT2, SNAT2, CAT1 and LAT2 involved in the uptake of neutral, branched-chain and aromatic aminoacids. Here, the overexpression of the *ASPG* gene, codying for the enzyme asparaginase, promoted apoptosis resistance due to the increased production of asparagine and glutamate elicited by NRF2–ATF4 activation, while the AKT inhibition sensitized NSCLC tumors to L-asparaginase depletion from the extracellular space [98]. Finally, in a very recent study focused on a model of NSCLC, the same group proved that NRF2 was required for the *KRAS*-dependent induction of the ATF4 pathway induced by nutrient withdrawal via PI3K/AKT signaling. Here, the concomitant presence of *KEAP1* loss-of-function (LOF) mutations promoted apoptosis upon ATF4 activation under nutrient stress. It is noteworthy that glutamine deprivation not only enhanced the ATF4-dependent transactivation of amino acid transporter genes and their protein expressions (LAT1, BCAT1, BCAT2), improving leucine and glutamine uptake, but also induced asparagine biosynthesis. Pharmacologic or genetic inhibition of the KRAS–NRF2–ATF4 axis exerted onco-suppressive effects both in vitro and in vivo suggesting that this pathway might be an attractive target for anticancer treatment [98]. Notably, similar observations were also reported in other types of tumors. For example, it has been shown that NRF2 can promote the ATF4 transcriptional activity in autophagy-deficient HCT116 CRC cells by disrupting its interaction with SIRT6 and promoting the expression of genes (*SLC6A9*, *SLC36A4*, *SLC38A1* and *SLC38A3*) codying for AATs (aminoacid transporters) involved in the import of alanine, proline, tryptophan, glycin and glutamine. Is it noteworthy that the AATs inhibition was able to enhance apoptosis in autophagy-deficient but not wild-type CRC cells upon glutamine withdrawal, revealing a tumor-specific vulnerability [99]. Interestingly, Guo et al. recently uncovered a previously uncharacterized posttranslational modification of NRF2 and its role in serine de novo synthesis and tumorigenesis in HepG2 and SMMC-7721 HCC cells. Here, NRF2 SUMOylation at lysine 110 (K110), was found to be required for the induction of ROS detoxification through the expression of *GPX2* and the subsequent increase in the levels of PHGDH, enhancing the tumorigenesis of HCC cells in vitro and in vivo. It should be noted that these changes stimulated the production of serine and one-carbon units required for purines synthesis, conferring resistance to both OS and serine starvation, two common conditions faced by cancer cells during malignant progression [98].

#### 3.3.2. NRF2 Controls xCT Antiport to Support Cell Survival Leading to Metabolic Addiction

Other work has focused on xCT, a transmembrane antiporter coded by the *SLC7A11* gene, which is often overexpressed in tumors and mediates the extrusion of glutamate in exchange of cysteine disulfide (Cys-SS) to refill the intracellular cysteine pool and support the redox balance [100,101,102]. In this regard, earlier studies have shown that NRF2 and ATF4 upregulation in T24 gallbladder cancer (GBC) cells can increase the xCT mRNA and protein levels, causing resistance to proteasome inhibition [103]. In another study, NRF2 was found to increase the expression of the *SLC7A11* gene and the activity of the xCT antiporter in MCF-7 BCC subdued to OS, while these changes were abrogated by *KEAP1* overexpression and mimicked by *KEAP1* silencing [104]. Interestingly, by screening a large dataset from almost 950 cancer cell lines, Shine et al. evidenced a positive correlation between the *NRF2* and the *SLC7A11* levels, especially within a subset of BCC. Further investigations revealed that *NRF2* silencing suppressed both xCT expression and glutamate export in Hs578T and MDAMB-231 BCC, conferring resistance to glucose deprivation due to enhanced mitochondrial respiration, while DMF-dependent NRF2 induction reverted this phenotype and promoted glucose addiction [105]. Consistently, Koppula et al. reported that glucose starvation induced *SLC7A11* expression in UMRC6 renal cancer cells through NRF2 and ATF4-dependent transcription, increasing their glucose dependence for cell survival. Indeed, *NRF2* or *ATF4* silencing attenuated the toxic effects of glucose withdrawal while the overexpression of *SLC7A11* reverted this phenotype and sensitized the UMRC6 cells to cell death induced by glucose removal [106]. Along similar lines, Sayin et al. have shown that the *KEAP1* LOF mutations can reduce nutrient flexibility and induce glutamine addiction in *KRAS*-driven lung adenocarcinoma cells of murine and human origin. Mechanistic insights revealed that increased xCT/*SLC7A11* expression prompted by NRF2 activation caused defects in the TCA cycle and glutamine anaplerosis due to enhanced glutamate extrusion. It is noteworthy that CB-839 dependent glutaminase inhibition, in the genetic context of *KEAP1* mutation, suppressed tumor growth in a panel of human cancer cells with different origin, while NRF2 activation by KI696 treatment sensitized *KEAP1*-WT cells previously refractory to CB-839. In this context, the authors proposed that the use of glutaminase inhibitors alone or in combination with NRF2 inducers might be a valid therapeutic strategy to target different cancers respectively carrying functional or mutated forms of the *KEAP1* gene [107]. It is important to notice that glutamine is a conditionally essential amino acid in many types of tumors that can be redirected towards different metabolic routes including the TCA cycle anaplerosis or the biosynthesis of hexosamines, aminoacids, nucleotides, fatty acids and GSH [108]. A quite recent study from the group of Savaskan revealed that NRF2 overexpression in patients affected by high-grade GBM resulted in a poorer clinical outcome and an overall reduced survival rate. By using F98 and U87 human glioma cells, the authors showed that the decreased *KEAP1* or enhanced *NRF2* expression fostered both cell proliferation and colony-forming activity, strongly increasing the mRNA and protein levels of the xCT antiporter and promoting resistance to ferroptosis. Importantly, the inhibition of xCT by erastin sensitized the F98 and U87 cells to ferroptosis induced by RSL3 treatment and promoted ROS accumulation—both these changes being exacerbated by *NRF2* silencing or *KEAP1* overexpression [109]. Additional work from the group of Kluza focused on melanoma cells with different sensitivity to the *BRAF* inhibitor Vemurafenib. Here, resistant cell lines (A375RIV1) displayed a marked increase in the NRF2 signaling resulting in the upregulation of *SLC7A11* and other genes involved in ROS scavenging (*GPX1* and *GPX2*), GSH synthesis (*GCLM*) and NADPH production (*TKT*, *TALDO1*) compared to the sensitive (A375-v) counterpart. It is noteworthy that *NRF2* silencing led to a marked decrease in the protein content of cytoprotective effectors in A375RIV1 cells and stimulated ROS accumulation, partially restoring their sensitivity to Vemurafenib [110]. Lastly, in a very recent work by LeBoeuf et al., *KRAS*-driven murine lung adenocarcinoma cells with LOF mutations of *KEAP1* displayed an enhanced antioxidant capacity and an altered metabolism, ultimately becoming highly dependent on the exogenous uptake of non-essential aminoacids (NEAAs) such as asparagine, glycine and serine, both in vitro and in vivo. Mechanistic insights revealed that these alterations could be phenocopied by pharmacologic NRF2 activation, that was causally linked to the depletion of the intracellular glutamate caused by xCT-mediated extrusion impairing NEAA synthesis and proliferation. It is noteworthy that the inhibition of glutaminase by CB-839 as well as the addition of oxidants was found to decrease the endogenous glutamate content and to sensitize cancer cells to NEAA withdrawal, even in absence of alterations in the KEAP1/NRF2 pathway [111]. These data suggest that interfering with the NRF2-xCT function can represent a valid therapeutic approach to overcome cancer cell resistance and promote ROS-dependent cytotoxicity. 

The potential link between NRF2 and glutamine metabolism in cancer cells has been confirmed by other work. For instance, an early study focused on HeLa cells to show that NRF2 could directly induce the expression of the *SLC1A5* gene codying for a glutamine importer [112]. More recently, NRF2 was seen to enhance the levels of the glutamine transporter coded by the *SLC1A4* gene and other metabolic enzymes in *KEAP1*-deficient esophageal squamous cancer cells (ESCC) of different origin inducing metabolic rewiring, while its genetic silencing or glycolysis inhibition decreased the ATP levels and the proliferation of human ESCC with high NRF2 levels [113]. Other data from Agyeman et al. suggested that NRF2 activation in MCF-10 and MCF-7 BCC due to *KEAP1* silencing or treatment with SFN caused the transactivation of the enzyme glutaminase [51]. Importantly, it has been proposed that the co-occurrence of *KEAP1* mutations with pre-existing oncogenic alterations can induce metabolic addicted phenotypes in cancer cells. In this regard, the group of Heymach took advantage of isogenic pairs of murine and human *KRAS*-driven lung adenocarcinoma cells (K tumors) with concomitant knockdown of *LKB1* (KL tumors) and *KEAP1* (KLK tumors) to explore the potential metabolic adaptations that might be targeted. Here, KL cells of murine and human origin displayed increased energetic stress and ROS accumulation, which was counteracted by the antioxidant response induced by the concomitant loss of *KEAP1*. It is noteworthy that KLK became glutamine addicted and CB-839 treatment suppressed the cell proliferation, inducing alterations in the redox balance and energetic stress, a phenotype partially rescued by the supplementation of adenine and totally reverted by pyruvate or glutamate repletion [114]. Finally, a very recent work exploring the role of NRF2 in *KRAS*-driven pancreatic ductal adenocarcinoma (PDAC) tumors, showed that the high levels of NRF2 identified cancers with poorer clinical outcomes characterized by increased resistance to gemcitabine and a marked dependence on glutamine metabolism and cystine uptake. It is noteworthy that either stable *NRF2* knockdown or glutaminolysis inhibition with CB-839 markedly enhanced the sensitivity of PDAC cells to gemcitabine, and the anticancer effect was further potentiated when CB839 was used in combination with gemcitabine in in vivo experiments [115]. Taken together, these data have indicated that NRF2 actively participates in the regulation of glutamine metabolism in malignant tumors suggesting that alterations in the KEAP1/NRF2 signaling alone or in combination with concomitant oncogenic activation might uncover specific metabolic vulnerabilities that might be therapeutically targeted to treat otherwise resistant tumors [116]. 

#### 3.3.3. NRF2 Regulates Cysteine Biosynthesis and Metabolic Transformation

Other lines of investigation have focused on cysteine, which is a limiting substrate for GSH biosynthesis and is therefore required to support the antioxidant systems of cancer cells. For this reason, malignant cells need to constantly replenish the intracellular pool of cysteine by promoting transporter-mediated uptake or de novo synthesis from methionine via the transsulfuration pathway. In this regard, it has been proposed that the activation of the transsulfuration pathway might render the cancer cells less dependent to the xCT antiporter and therefore confer resistance to ferroptosis. Indeed, Liu et al. showed that the prolonged inhibition of the xCT antiporter with erastin induced resistance to ferroptosis in SKOV3 and OVCA429 ovarian cancer cells. Mechanistically, constitutive NRF2 activation was found to play a causative role through the transcriptional upregulation of *CBS* (cystathione-beta-synthase), an enzyme that catalyzes the rate-limiting step in the transsulfuration pathway, while *NRF2* silencing or *CBS* knockdown were sufficient to enhance the OC cell susceptibility to ferroptosis through the induction of OS and lipid peroxidation [117]. Another study has suggested that cancer cells can also preserve the intracellular cysteine content by suppressing its conversion into other unnecessary metabolites. In this regard, Kang et al. have recently shown that NRF2 plays a crucial role in promoting the accumulation of cysteine and its funneling into multiple downstream pathways in lung cancer cells. Here, the analysis of TCGA data from NSCLC patient samples revealed that CDO1 (cysteine dioxygenase 1), an enzyme that converts cysteine (CYS) to cysteine sulfinic acid (CSA), was epigenetically silenced by promoter methylation and this change predicted a poor prognosis. Consistently, NRF2 expression in NSCLC cells stimulated the uptake of cystine via xCT and thereby CYS accumulation, while the forced restoration of CDO1 conversely depleted the intracellular pool of CYS favoring its metabolic conversion to CSA, which was subsequently extruded or transformed into sulfites. In turn, the constant reduction of cystine to cysteine caused NADPH depletion, increasing the sensitivity of NRF2-expressing cells to lipid peroxidation and impairing NADPH-linked biosynthesis required for cell proliferation [118]. These data have indicated that cancer cells can exploit NRF2 activation to preserve the intracellular pool of cysteine by preventing its rerouting into futile metabolic pathways. On the other hand, CDO1 might represent a metabolic liability in lung cancers with high intracellular levels of cysteine, especially in the context of NRF2 overactivation that is expected to generate a nutritional phenotype susceptible to therapeutic strategies targeting this vulnerability. 

It should be noticed that the transsulfuration pathway is constitutively activated in several cancer cell lines, wherein it promotes de novo cysteine synthesis to support redox homeostasis and tumor growth in vivo. Despite the fact that the role of NRF2 still needs to be further elucidated in this context, it is expected that this topic will become the focus of extensive research in the near future, representing a crucial regulatory node at the intersection between metabolic reprogramming and redox homeostasis [119]. In conclusion, it is becoming increasingly clear that oncogene-induced metabolic alterations driving malignant progression might in turn represent unfavorable events under specific nutritional conditions, wherein the limiting availability of a specific metabolite can be further exacerbated by targeting key regulators responsible for tumor adaptation to induce cancer cells demise. 

### 3.4. NRF2 Is Regulated by H2S Metabolism

In recent years, the gasotransmitter hydrogen sulfide (H_2_S) has emerged as an important mediator of tumor biology implicated in the regulation of cell proliferation, bioenergetics, migration, invasion and tumor angiogenesis with pro- and anti-oncogenic effects [120,121]. Importantly, the use of H_2_S modulators is considered a promising anti-cancer strategy in different types of tumors [122] while a potential crosstalk with NRF2 signaling is suggested by a number of studies. In this respect, earlier evidence has suggested that H_2_S can induce NRF2 nuclear accumulation and protect cardiomyocytes against ischemia [123], while later studies showed that H_2_S could inactivate KEAP1 by promoting the formation of an intramolecular disulfide bond (C226–C613) and thereby induce NRF2 stabilization, at least in part through H_2_O_2_ generation. Moreover, H_2_S was seen to induce the NRF2-dependent overexpression of *Cbs* (Cystathionine-β-synthase) and *Cse* (Cystathionine gamma-lyase) in MEFs, two enzymes involved in the biosynthesis of cysteine and H_2_S itself, revealing the existence of a reciprocal interrelation between these two pathways [124]. Accordingly, Yang et al. showed that MEFs isolated from *Cse*-KO mice displayed signs of increased OS and accelerated cellular senescence compared to MEFs-WT, while the administration of an H_2_S donor was able to rescue this phenotype promoting KEAP1 inactivation (C151-S-sulfhydration), Nrf2 nuclear translocation and increased GSH synthesis [125]. In agreement with these observations, H_2_S was also found to attenuate atherosclerosis in a mouse model of diabetes, promoting Keap1 S-sulfhydration at Cys 151 and the NRF2-dependent transactivation of antioxidant enzymes [126]. However, Koike et al. first proved that H_2_S can also activate the NRF2 pathway in cancer cells, since the polysulfide N_2_S_4_, a product of H_2_S signaling, was found to protect Neuro2A NB cells from the cytotoxic effects of tert-butyl hydroperoxide (t-BHP), promoting an increased GSH synthesis and HO-1 expression due to *KEAP1* inactivation and enhanced NRF2 nuclear translocation [127]. Further validation came from a more recent study wherein the exogenous administration of an H_2_S donor (NaHS) was seen to protect SH-SY5Y NB cells from ischemia–reperfusion injury induced by glucose deprivation/reoxygenation, through the increased expression of NRF2, ERK and p38MAPK [128]. Interestingly, recent work has suggested that H_2_S might exert both pro-oncogenic and anticancer effects. Indeed, Shan et al. proved that the administration of diallyl disulfide (DADS), a slow H_2_S donor, was able to dose-dependently attenuate the incidence of skin cancer in a mouse model of chemically-induced carcinogenesis, by promoting p21/NRF2 interaction and the upregulation of several antioxidant enzymes due to enhanced NRF2 nuclear accumulation [129]. In accordance, by using a model of human gastric cancer, Jiang et al. proved that the exposure of BGC-823 cells to the H_2_S donor DATS (diallyl trisulfide), was able to impair NRF2 and AKT activation inducing cell cycle arrest and cell demise through the modulation of BCL-2 family proteins involved in the initiation of mitochondrial-dependent apoptosis. It is noteworthy that the DATS treatment of xenografted mice reduced both the tumor volume and weight compared to the control mice, also potentiating the anticancer efficacy of cisplatin by suppressing the NRF2/AKT pathways [130]. In marked contrast however, Wang et al. recently proved that H_2_S can promote NRF2 nuclear translocation and therefore upregulate the levels and the activation of CD36, a receptor involved in the uptake of fatty acids, ultimately promoting metastasis formation in a model of gastric cancer [131]. These data have suggested that the H2S/NRF2 axis has profound implications in the development and clinical management of cancer, while the outcome of its activation towards pro-oncogenic or anti oncogenic roles might be dictated by the individual context represented by the presence of specific regulators and their functional interrelation in tumors of different origin. 

### 3.5. NRF2 Controls Iron Metabolism 

Another key function of the NRF2/KEAP1 pathway is the regulation of iron metabolism. Consistently with its antioxidant role, NRF2 not only controls the intracellular levels of heme through the HO-1 enzyme but also the removal of iron atoms derived from its catalytic activity by promoting the expression of proteins involved in iron storage or export such as ferritin and ferroportin, preventing the engagement of free iron into ROS-producing reactions (i.e., Fenton and Haber–Weiss). Notably, NRF2 can regulate the expression of two key enzymes: biliverdin reductase, that converts biliverdin into the antioxidant molecule bilirubin and ferrochelatase, which is involved in the synthesis of heme, an important cofactor for catalase and a substrate for bilirubin generation [132]. In the context of tumor development, it is widely accepted that an excess of iron can induce carcinogenesis due to increased OS, while perturbations in the iron metabolism can promote tumor progression and metastasis formation modulating cancer cell survival and adaptation to the microenvironment. On the other hand, cancer cells also need to prevent the potentially harmful effects of iron overload that might otherwise trigger ferroptosis, an iron-dependent non-apoptotic form of cell death [133]. Earlier studies have shown that NRF2 can induce the expression of the iron-storage protein ferritin H in murine Hepa1–6 hepatoma, human NIH3T3 fibroblasts and HepG2 HCC cells treated with dithiolethiones [134], while NRF2 was recently found to constitutively bind the AREs of the ferritin H gene in HepG2 cells, probably to ensure its basal expression [135]. Further mechanistic insights from the group of Mukhtar revealed that the overexpression of HO-1 driven by constitutive NRF2 activation was able to increase the resistance of A549 NSCLC cells to the proxidant EGCG, a phenotype reverted by *HO-1* silencing, as well as the inhibition of NRF2 nuclear translocation and iron chelation by Desferoxamine (DFO) [136]. Additional work has shown that the intracellular content of the iron exporter ferroportin (FPN) was markedly downregulated in MDA-MB-231 BCC and associated with enhanced tumor growth both in vitro and in vivo, reverted by forced FPN expression. Mechanistically, NRF2 and MZF-1 were found to directly induce *FPN* gene expression and their content was markedly reduced in tumor specimens from BC patients compared to normal tissues, indicating that NRF2 can drive tumorigenesis by also restricting iron egression through decreased FPN induction [137]. Consistently, a later study confirmed that NRF2 and FPN might exert an onco-suppressive role in certain tumors, since PC3, DU145 and LNCAP PC cells were found to express low mRNA levels of both these proteins, while forced NRF2-dependet FPN overexpression strongly abrogated the migration, mitosis and the survival of PC3 cells [138]. Importantly, recent evidence has suggested that NRF2 is a key determinant of cancer cell sensitivity towards ferroptosis inducers or common anticancer drugs. In this respect, Sun et al. first showed that the overactivation of NRF2 signaling prompted by p62-mediated KEAP1 degradation was causally linked to ferroptosis resistance in Hepa 1–6 and HepG2 cells both in vitro and in vivo. It is noteworthy that the genetic or pharmacologic inhibition of NRF2, as well as the knockdown of its downstream target genes *NQO-1*, *HMOX-1* and *FTH1* was sufficient to induce growth inhibition and increase the anticancer effects of erastin and cisplatin [139]. In partial contrast, however, it has also been reported that under certain circumstances NRF2-dependent HO-1 induction can promote rather than attenuate ferroptosis and OS. Indeed, in a recent work aimed at elucidating the anticancer effects of the NF-kB inhibitor BAY 11-7085 (BAY), this compound was found to induce ferroptosis in human cancer cell lines of different origin, which was NF-kB independent but largely dependent on iron overload and altered redox homeostasis. Here, BAY increased the protein levels of NRF2 and its nuclear translocation presumably through KEAP1 downregulation, leading to the induction of cytoprotective target genes. Among them, HO-1 also accumulated within the nucleus and unexpectedly promoted iron overload, mitochondrial damage, OS and cell death, which were attenuated by the high expression of the xCT antiport but conversely exacerbated by the xCT inhibition through erastin [140]. Intriguingly, non-canonical NRF2 activation in tumors can also derive from the intracellular accumulation of oncometabolites caused by metabolic reprogramming. Indeed, as evidenced by recent work, increased levels of fumarate, due to genetic deficiency of fumarate hydratase, were found to upregulate the expression of *FTL* and *FTH* genes of ferritin subunits, respectively promoting mRNA translation through IRP2 inactivation and NRF2-dependent genetic induction in UOK262 renal cell carcinoma cells [141]. Importantly, by regulating the iron content NRF2 can also induce chemoresistance, as emphasized by recent studies. For instance, by comparing A2780, COC1 and PEO1 OVAC cells, with different sensitivities to cisplatin, Wu et al. have shown that the resistant counterparts were characterized by elevated *NRF2* mRNA levels and a reduced mRNA content of *SLC40A1* (Solute carrier family 40 member 1) codying for FPN. Mechanistically, NRF2 was found to decrease *SLC40A1* expression and cisplatin sensitivity by preserving the intracellular iron content. Indeed, the iron chelator desferal was able to overcome the chemoresistant phenotype and its efficacy was further potentiated when used in combination with the NRF2 inhibitor brusatol (BR) [142]. Similarly, the anti-malarian artesunate was recently found to induce suboptimal ferroptosis in cisplatin-resistant head-neck cancer (HNC) cells. Here, artesunate prompted NRF2-dependent HO-1 upregulation and increased the ferroptosis resistance, while genetic or pharmacologic disruption of NRF2 markedly sensitized chemoresistant HNC cells to artesunate both in vitro and in vivo [143]. As evidenced by Campbell et al., the list of NRF2 target genes codying for proteins involved in iron and heme metabolism, including *FTL*, *FTH1*, *AMBP*, *ABCB6*, *FECH*, *HRG-1* (*SLC48A1*) and *TBXAS1*, is constantly expanding but their functional role in cancer-specific contexts still needs to be fully elucidated [144]. Taken together, these data have indicated that targeting the NRF2-dependent regulation of iron metabolism might be a valid strategy to combat chemoresistance and induce ferroptosis in a variety of malignant tumors, especially in those forms that are refractory to apoptosis inducers by virtue of intrinsic or acquired resistance caused by defects in the apoptotic machinery. 

### 3.6. NRF2 Controls Redox Homeostasis through NADPH Synthesis

Accumulating evidence has indicated that cancer cells are characterized by increased steady-state levels of ROS caused by epigenetic, oncogenic and metabolic alterations. On the other hand, this selective pressure requires an increased efficiency of the antioxidant systems that constantly transform, remove and neutralize ROS molecules and their effects to concomitantly support redox signaling and prevent excessive damage to the biomolecules. In this regard, the aberrant activation of NRF2 in cancer cells is frequently associated with the overexpression of enzymes controlling the redox status through direct or indirect mechanisms [145,146,147,148,149]. Importantly, the coenzyme NADPH is rapidly emerging as a key regulator of cancer cell antioxidant systems acting as an electron donor for at least two different redox nodes and ROS-scavenging enzymes. The first is represented by the glutathione/glutaredoxins system relying on the NADPH-dependent regeneration of GSH (reduced form) from its oxidized counterpart (GSSG) catalyzed by the glutathione reductase (GR) [150]. The second includes the thioredoxin reductase (TRXR) that restores the reduced form of thioredoxins (TRXs) at the expense of reducing equivalents derived from the NADPH and indirectly supports the thioredoxin/peroxiredoxin system involved in the removal of H_2_O_2_ and the reduction of protein thiol groups [151,152]. Lastly, the enzymes’ glutathione peroxidases (GPXs), implicated in the reduction of H_2_O_2_ or lipid hydroperoxides, require the reducing power of GSH for their regeneration and are therefore linked to NADPH through the GR activity. It is noteworthy that this cofactor is mainly regenerated by a group of NADP^+^-dependent metabolic enzymes belonging to the pentose phosphate pathway (PPP), TCA or folate cycle that under specific contexts can be directly or indirectly modulated by NRF2 activation. Indeed, in a seminal paper from Mituishi et al., the integrated analysis of ChIP-seq and microarray data on A549, EBC-1, H2126 and LK-2 lung cancer cells revealed that NRF2 could directly induce genes codying for NADPH-producing enzymes such as *G6PD*, *PGD* (phosphogluconate dehydrogenase), *TKT* (transketolase), *TALDO1* (transaldolase 1), *ME1* and *IDH1*. It is noteworthy that the increased expression of PPP enzymes accounted for enhanced tumor growth both in vitro and in vivo while the constitutive activation of the PI3K/AKT signaling was found to further increase the NRF2-dependent transcription [153]. Accordingly, the stable knockdown of *KEAP1* was also found to enhance while *NRF2* silencing repressed the expression of *TKT* in HCC cells, wherein this enzyme promoted NADPH synthesis, tumor growth, metastasis formation and sorafenib resistance [154]. In another study focused on A172 and U87 glioma cells, the hTERT (human telomerase) was found to act as a downstream effector of NRF2 activation required for the upregulation of *G6PD* and *TKT* genes, suggesting the existence of a regulatory loop that might be active in tumor specific contexts and mediate the metabolic adaptation of cancer cells [155]. Other work showed that NRF2 overexpression and *KEAP1* knockdown increased, while conversely the *KEAP1* overexpression and *NRF2* silencing reduced the expression level of G6PD and TKT in MCF-7 and MDA-MB-231 BCC both at the mRNA and protein levels. Despite the fact that the potential effects on NADPH levels were not investigated, NRF2 activated the G6PD/HIF-1α axis to enhance proliferation and migration of BCC through increased EMT (epithelial–mesenchymal transition) suggesting potential NADPH uprising [156]. Interestingly, Singh et al. revealed the existence of alternative mechanisms of NRF2-driven PPP gene induction in DU145 PC and A549, H1437 NSCLC cells. Here, the sustained activation of NRF2 due to *KEAP1* LOF mutation was seen to promote tumor growth and upregulate the expression of PPP genes, including *G6PD*, *TKT* and *PGD*, through the epigenetic inhibition of miR-1 and miR-206, two repressors of metabolic gene induction. Mechanistically, *NRF2* silencing reduced the histone deacetylase HDAC4 (histone deacetylase 4) but conversely enhanced both miR-1 and miR-206 levels, while their overexpression abrogated metabolic genes induction, NADPH generation, ribose synthesis and tumor growth in vivo [157]. The functional link between NRF2 activation and miRNA repression has been also substantiated in a rat model of hepatocarcinogenesis, wherein downregulated miR-1 expression was observed in nodules characterized by high levels of NRF2 target genes expression, including *G6PD*. Here, *NRF2* silencing blunted G6PD expression and conversely increased miR-1 levels, while opposite changes were induced by miR-1 mimics. A further analysis on a cohort of 59 HCC patients revealed that the association between high levels of *G6PD* mRNA and low miR-1 expression was positively correlated with grading, metastasis formation and poor prognosis. It is noteworthy that despite the fact that the potential role of NRF2 was not strictly proved in this analysis, high mRNA levels of its target gene *NQO1* were observed in the cohort of HCC patients, suggesting NRF2 activation [158].

Lastly, in a quite recent study, combinatorial CRISPR-Cas9 screens coupled with a metabolic fluxes analysis were performed on HeLa and A549 cancer cells to investigate the potential interactions and dispensability of metabolic genes regulating glycolysis and PPP. In this regard, either loss or inactivating mutations of *KEAP1* were found to upregulate the NRF2-dependent expression of genes involved in the GSH synthesis and NADPH regeneration, including *G6PD* and *PGD*, despite the different cell lines exhibiting distinct dependency and compensatory mechanisms to cope with defective PPP function caused by the genetic KO of these enzymes or the forced expression of *KEAP1* WT. Therefore the authors proposed that the integration of genetic screening and functional metabolic flux analysis might serve to decipher the context wherein targeting metabolic alterations induced by NRF2 could produce the most beneficial therapeutic effects in selected patients cohorts [159]. Notably, another work suggested that NRF2 could also induce the expression of TCA cycle enzymes involved in NADPH synthesis, such as *ME1* and *IDH1* [153,160] or that of the folate cycle enzyme *MTHFDL1* (methylenetetrahydrofolate dehydrogenase 1-like), that has been shown to enhance cell proliferation and sorafenib resistance in HCC both in vitro and in vivo through metabolic rewiring and increased NADPH generation [161], in agreement with other data [162]. In conclusion, these studies have demonstrated or suggested that interfering with NRF2-driven NADPH generation can potentially impair tumor growth and survival affecting both anabolic processes and redox homeostasis, representing a promising therapeutic option against a number of different cancers. A schematic illustration of the metabolic pathways influenced by NRF2 is shown in Figure 3.

## 4. Conclusions

NRF2 was first recognized in anticancer research as an inducer of several antioxidant enzymes. It can protect cells and tissues against many types of toxicants and carcinogens by increasing the expression of cytoprotective genes. Over the last decade, studies showed that there are beneficial and detrimental roles of NRF2. Under the normal conditions, NRF2 works as a defender of oxidative stress damage. However, for the cancer patients this might be harmful and associated with both tumor progression and therapy resistance. These two paradoxical aspects have been defined as the ‘dual role of NRF2′ [157,158]. In agreement with this view, it is increasingly recognized that several NRF2 activators possess both electrophilic and pro-oxidizing properties and do not show a simple dose–response relationship, but rather a U-shaped profile, which is consistent with a hormetic behavior. Hence, the identification of the dose associated with the beneficial effects of NRF2 activation, especially in vivo, would require a full understanding of the underlying mechanisms and the specific context of its activation (i.e., genetic background, age, gender, role of other regulators) to avoid undesired off-target effects.

The presently described NRF2 inhibitors have been stated to suppress the NRF2 pathway through different mechanisms and contexts [159]. The level of NRF2 is kept low in normal cells, however very high in cancer provoking therapy resistance. It is now recognized that NRF2 plays important roles in apoptosis, cell cycle progression and stem cell differentiation. These activities could be related to the presence of multiple binding proteins [160]. Higher NRF2 levels have been reported in different cancer tissues, such as lung [161], pancreas [162] and endometrium [163]. Several studies proved that somatic KEAP1 mutations are present in the tumor tissue of the lung [164], liver [165] and ovarian cancers [166]. The mechanism of sustained NRF2 activation has been revealed in some hereditary cancer types [167] and is related to the effects on genes coding for metabolic enzymes which can in turn affect key cysteine residues on KEAP1 to disrupt NRF2 interaction. In hepatocellular carcinoma, genomic modifications were mainly detected in the KEAP1 gene [168] and in urothelial bladder carcinoma a probable link with NRF2 and thioredoxin signaling was observed [169].

NRF2 plays a central role in drug resistance in several patients undergoing chemotherapy wherein NRF2 overexpression frequently reduced the sensitivity to the anti-cancer compounds [170]. Based on this assumption, NRF2 inhibition is a possible beneficial approach in cancer treatment. NRF2 targeting by RNAi helps to enhance cancer cell sensitivity to several anti-cancer drugs [171]. The action of NRF2 in cancer is very multifaceted. Indeed, several papers have shown that NRF2 has a significant role in preventing tumor development. In this regard an exemplary case is represented by hematologic tumors. Indeed, some studies have reported that the modulation of NRF2 activity can exert anti-tumor effects in different leukemic cells. For instance, it has been shown that some NRF2 activators can synergistically potentiate the pro-differentiating effects induced by vitamin D derivatives in a pre-clinical mouse model of acute myeloid leukemia (AML) [163]. On the other hand, the NRF2-driven overexpression of aldo keto reductase 1C (AKR1C1) enzymes has been associated with therapy resistance in T-cell acute lymphoblastic leukemia (T-ALL) and their genetic or pharmacologic inhibition to enhance the efficacy of vincristine treatment [164]. Similarly, NRF2 overexpression in AML cells was seen to cause insensitivity towards cytarabine and daunorubicin, two common anticancer drugs used in this type of leukemia. Here, the genetic silencing of NRF2 or its enhanced degradation caused by treatment with brusatol markedly reverted the chemoresistant phenotype of the THP1 cells and promoted ROS-dependent apoptosis [165]. This clearly indicates that the potential therapeutic benefits derived from NRF2 modulation are not limited to solid tumors but include also a variety of hematologic malignancies.

Although the role of NRF2 in cancer is still debated, many studies have revealed that NRF2 knockout animal models are prone to chemically induced carcinogenesis. This finding suggests that NRF2 acts as a probable tumor suppressor against carcinogenesis [172]. In contrast, NRF2 is overexpressed in several types of cancer cells, wherein it confers a survival advantage towards adverse conditions, including therapeutic treatments [173]. Overall, compelling research suggests a protective role of NRF2 especially in the early phases of cancer development, however in later stages, NRF2 overexpression supports cancer cells to adapt to the microenvironment [174]. Hence, it is expected that NRF2 inhibitors would sensitize tumor cells to anticancer treatments and open new avenues in the fight against cancer.

At this point, many of the existing NRF2 activators and inhibitors have been shown to target other biological effectors that are however implicated in the regulation of the NRF2 pathway. In this regard, additional compounds targeting effectors of NRF2 signaling should be designed, identified and tested in selective clinical trials. Many novel strategies such as the targeting of β-TrCP-NRF2, the HRD1–NRF2 binding or developing PROTACs (proteolysis-targeting chimeras) to promote NRF2 proteasomal degradation, should be examined to expand the armamentarium to be used in cancer prevention and therapy.

## Figures and Tables

**Figure 1 biomolecules-10-00791-f001:**
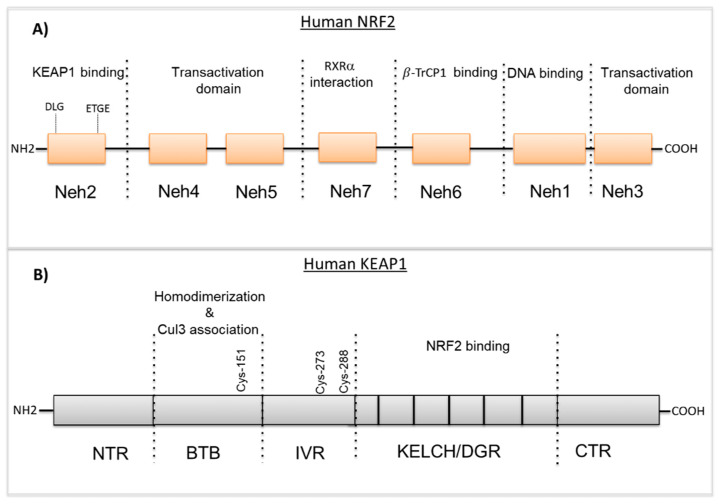
The structures of NRF2 and KEAP1. (**A**) NRF2 includes seven highly conserved domains Neh1–Neh7. The N-terminal Neh2 domain contains DLG and ETGE motifs that control the KEAP1 interaction. The Neh3, Neh4, and Neh5 domains are called transactivation domains. The Neh7 domain is required for the interaction with the Retinoid X receptor-α. The Neh6 domain is a serine-rich domain that binds to β-TrCP. The C-terminal domain, Neh1, is responsible for DNA-binding and hetero-dimerization with small MAF proteins (sMAFs). (**B**) KEAP1 contains five domains. The BTB domain mediates KEAP1 homodimerization and Cul3-E3-ligase binding. The IVR contains critical reactive cysteine residues that are essential for controlling the NRF2 activity. The Kelch/DGR domain is required for interaction with the Neh2 domain of NRF2. BTB, broad complex, tram-track and bric-a-brac; CTR, C-terminal region; Cul3, Cullin3; IVR, intervening region; KEAP1, kelch-like ECH-associated protein 1; sMAFs, musculoaponeurotic fibrosarcoma oncogene; Neh, NRF2–ECH homologous structure; NRF2, nuclear factor erythroid-2-related factor-2; NTR, N-terminal region; RXRα.

**Figure 2 biomolecules-10-00791-f002:**
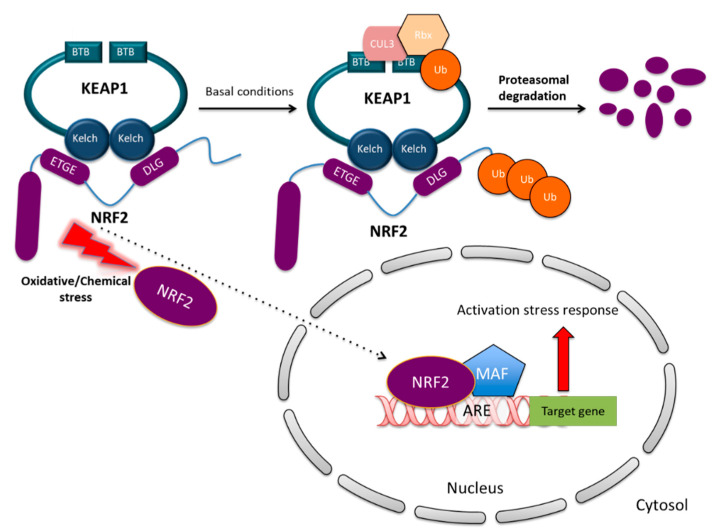
NRF2 regulation through KEAP1 under basal and stress conditions. Under basal conditions, NRF2 localizes in the cytosol and interacts with the Cul3-Rbx1 E3 ubiquitin-ligase substrate KEAP1 that constantly primes NRF2 for ubiquitination and proteasomal degradation. Oxidative/electrophilic stress causes conformational changes of KEAP1 through the modification of cysteine residues in IVR and BTB domains leading to NRF2 dissociation. Free NRF2 enters into the nucleus where it forms dimers with small MAF proteins or other interactors and binds to the antioxidant responsive elements (AREs) regulatory sequences of target genes, inducing their expression.

**Figure 3 biomolecules-10-00791-f003:**
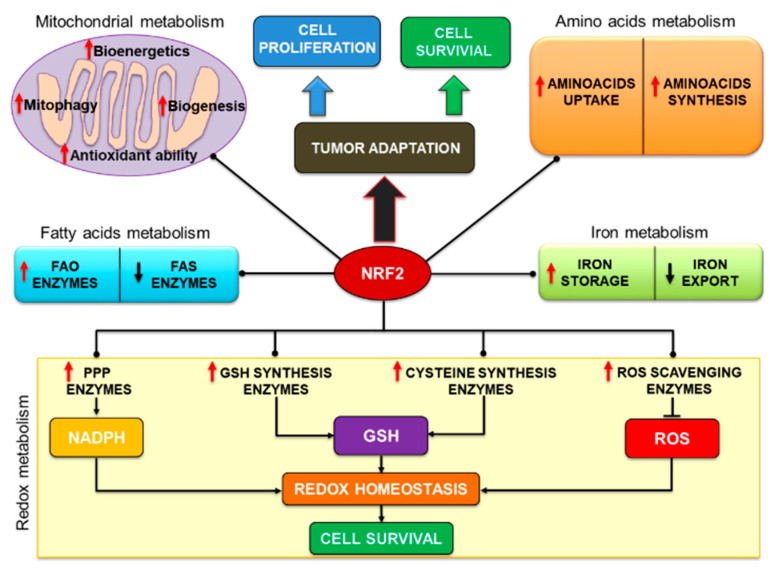
NRF2 regulates the multiple metabolic pathways of malignant cells. NRF2 transcriptionally controls the expression of genes controlling the expression of direct or indirect modulators of cancer metabolism, ultimately promoting tumor adaptation, cell proliferation and cell survival. List of abbreviations: FAO: Fatty acids oxidation; FAS: Fatty acids synthesis; NADPH: Reduced form of nicotinamide adenine dinucleotide phosphate; GSH: Reduced form of glutathione; ROS: Reactive oxygen species.

**Table 1 biomolecules-10-00791-t001:** A list of selected genes regulated by NRF2.

Biochemical Function	Gene Symbol	Extended Name	References Number
**Antioxidants**	*GCLC*	Glutamate–cysteine ligase, catalytic subunit	[30,50]
	*GCLM*	Glutamate–cysteine ligase, modifier subunit	[25,45]
	*GPX1,2,4*	Glutathione peroxidase 1	[25]
	*GSR1*	Glutathione reductase 1	[51]
	*NQO1*	NAD(P)H:quinoneo xidoreductase 1	[51]
	*SLC7A11*	Sodium-independent cysteine-glutamate antiporter	[52]
	*SRXN1*	Sulfiredoxin 1	[52]
	*PRDX1*	Peroxiredoxin 1	[53]
	*TXN1*	Thioredoxin	[28,30]
	*TXNRD1*	Thioredoxin reductase 1	[30]
**Phase I detoxification**	*ADH7*	Alcohol dehydrogenase class 4 mu/sigma chain	[40]
	*AKR1B1,*	Aldo-keto reductase family 1 member B1	[23]
	*AKR1B8*	Aldo-keto reductase family 1. member B8	[23]
	*AKR1B10*	Aldo-keto reductase family 1. member B10	[23]
	*AKR1CL*	Aldo-keto reductase family 1. member C-like	[23]
	*ALDH1A1*	Aldehyde dehydrogenase 1 family member A1	[54]
	*ALDH3A1*	Aldehyde dehydrogenase 3 family member A1	[23]
	*CBR1*	Carbonyl reductase 1	[46]
	*CYP1B1*	Cytochrome P450	[40]
	*PTGR1*	Prostaglandin reductase 1	[40]
	*EPXH1*	Epoxide hydrolase 1, microsomal	[40]
**Phase II detoxification**	*GSTA1,2*	Glutathione S-transferase alpha 1,253,4	[26]
	*GSTM1,2,3,4*	Glutathione S-transferase mu 1	[26]
	*MGST1*	Microsomal glutathione S-transferase	[55]
	*UGT1A1*	UDP Glucuronosyltransferase 1	[46]
	*UGT1A2*	UDP glucuronosyltransferase 1 family. polypeptide A2	[46]
**Phase III detoxification**	*ABCB6*	ATP-binding cassette, subfamily B (MDR/Tap) member 6	[56]
	*ABCC1*	ATP-binding cassette, subfamily C(CFTR/MRP)	[34]
	*ABCC2*	ATP-binding cassette, subfamily C(CFTR/MRP)	[34]
	*ABCC3*	ATP-binding cassette, subfamily C(CFTR/MRP)	[34]
	*ABCC4*	ATP-binding cassette, subfamily C(CFTR/MRP)	[34]
	*ABCC5*	ATP-binding cassette, subfamily C(CFTR/MRP)	[33]
**Heme and iron metabolism**	*FTH1*	Ferritin heavy chain 1	[48]
	*FTL1*	Ferritin light chain 1	[46]
	*HMOX1*	Heme oxygenase 1	[32]
**NADPH generation**	*G6PD*	Glucose-6-phosphate dehydrogenase	[31]
	*IDH1*	NADP-dependent isocitrate dehydrogenase	[31]
	*PGD*	6-phosphogluconate dehydrogenase	[31]
	*ME1*	Malic enzyme 1	[31]
**Apoptosis**	*BCL2*	B-cell lymphoma 2	[44]
**Autophagy**	*ATG5*	Autophagy protein 5	[48,57]
	*ATG7*	Autophagy protein 7	[48,57]
	*LC3B*	Microtubule-associated protein 1A/1B-light chain 3B	[43,52]
	*ULK1*	UNC-51 autophagy-activating kinase 1	[58]
**Proteasomal degradation and unfolded protein response**	*ATF4*	Activating transcription factor-4	[59,60]
	*PSMA1*	Proteasome subunit alpha type-1	[50]
	*PSMB5*	Proteasome subunit beta type-5	[50]
	*PSMC1*	Proteasome AAA-ATPase subunit Rpt2	[50]
	*SQSTM1*	Sequestosome 1 (p62)	[61]
**Regulation of NRF2 signaling**	*KEAP1*	Kelch-like ECH-associated protein 1	[62]
	*NFE2L2*	Nuclear factor, erythroid 2-like 2 (NRF2)	[63,64]

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
