# Peer review of "The NRF2/KEAP1 Axis in the Regulation of Tumor Metabolism: Mechanisms and Therapeutic Perspectives"

_biomolecules, 2020, doi:10.3390/biom10050791_

Round 1

Reviewer 1 Report

The authors have written a comprehensive, timely, review of the complex interaction between Nrf2, metabolic reprogramming and redox homeostasis, including valuable figures and a table. The major weakness of the review is the numerous grammatical errors found throughout the manuscript.

Other minor points:

Canonical Nrf2 activation is shown in Fig 2 and described in the text, but a few sentences and references should be added regarding non-canonical activation of the Nrf2 pathway.

Correlation is not proof of causation and it is not evident if activation of Nrf2 is the cause of or response to the described mitochondrial dysfunction and altered metabolism.

The summary sections at the end of each sections provide useful conclusions but do not need to be highlighted in gray.

Additional comments on the importance of dose response should be included, as the authors describe differences in modulation of mitochondrial dynamics by sulforaphane (lines 191-193), diallyl disulfide (lines 470 and beyond), and ferroptosis (lines 523-524).

Add a paragraph and references (PMID: 30610225) regarding the current status of Nrf2 inhibitors as this has been a challenging area of investigation. Is it possible that targeting the numerous downstream interactions will limit the value of such inhibitors, because of a lack of specificity or toxicity?

Author Response

We thank the reviewer for his/her suggestions and comments. We went through the main text and indeed found some, yet minor, grammatical errors that we have now amended. We hope that in this form the manuscript will be more appreciated by the readership of Biomolecules.

Concerning the minor points:

-A brief description of the major mechanisms of non-canonical regulation of NRF2 stability, with their relative references, has been implemented (see lines 107-113).

-Regarding the role of NRF2 in mitochondrial function, we agree that in some of the reported studies there is only a correlation but not a formal demonstration of its causal role. We have therefore decided to stress this aspect when necessary. (i.e. lines 192-193; 198-200; 215-216;224-225; 262-264)

-For what concerns the sections at the end of the subparagraphs, now the grey color has been removed.

-Following the reviewer’s suggestion we have now added in the “Conclusions” a comment on the dualistic effects exerted by NRF2 activation or inhibition, in agreement with the hormetic hypothesis of NRF2 function (see lines 655-660).

-Lastly, despite the paramount importance of the topic, we feel that discussing the general aspects of NRF2 inhibitors is a little bit beyond the scope of our review. However, we are planning to extensively cover the aspects related to NRF2 inhibition in cancer in a nextcoming manuscript currently under preparation.

Reviewer 2 Report

The Nrf2/Keap1 Axis in The Regulation of Tumor Metabolism: Mechanisms and Therapeutic Perspectives

Panieri er al.

The authors review Nrf2/Keap1 signaling in tumor metabolism. In general this is a very well-constructed, well written manuscript with a balanced level of detail. A number of Nrf2-Keap reviews, highlighting pro-oncogenic functions, have been published in the last few years. A couple of small points should be addressed:

The inducers of Nrf2 (pg 7) for example sulforaphane are considered only tool compounds where off target effects of should be considered.

The impacts of Nrf2 activity and inhibition on tumors of different origins (epithelial, mesodermal, hematopoietic) could be considered, for example contrasting carcinomas, sarcomas and hematologic malignances. Most or all of the data presented reflect epithelial derived carcinoma solid tumors. This also might differentiate from other reviews and would add a new dimension to the review.

Author Response

We are glad that the reviewer appreciated our manuscript and we thank the reviewer for his/her suggestions.

-We agree that it is important to underscore the occurrence of potential off targets effects associated with pharmacologic manipulation of NRF2. We have now added few sentences that will help the reader to have a more balanced view concerning benefits and drawbacks of NRF2 manipulation (see lines 201-204 and 661-664), in agreement with the concept of NRF2 modulation as a truly hormetic response.

-Following the reviewer’s suggestion we have now reported some examples wherein NRF2 activation or its inhibition in hematologic tumors has shown promising anti-tumor effects (see lines 682-695).